# A nonrandomized cohort and a randomized study of local control of large hepatocarcinoma by targeting intratumoral lactic acidosis

Ming Chao[1], Hao Wu[2], Kai Jin[1], Bin Li[1], Jianjun Wu[1], Guangqiang Zhang[1], Gong Yang[3], Xun Hu[2]*

[1]Department of Radiology, The Second Affiliated Hospital of Zhejiang University School of Medicine, Hangzhou, China; [2]Cancer Institute, The Second Affiliated Hospital of Zhejiang University School of Medicine, Hangzhou, China; [3]Vanderbilt University Medical Center, Nashville, United States

## Abstract

**Background:** Previous works suggested that neutralizing intratumoral lactic acidosis combined with glucose deprivation may deliver an effective approach to control tumor. We did a pilot clinical investigation, including a nonrandomized (57 patients with large HCC) and a randomized controlled (20 patients with large HCC) study.

**Methods:** The patients were treated with transarterial chemoembolization (TACE) with or without bicarbonate local infusion into tumor.

**Results:** In the nonrandomized controlled study, geometric mean of viable tumor residues (VTR) in TACE with bicarbonate was 6.4-fold lower than that in TACE without bicarbonate (7.1% [95% CI: 4.6%–10.9%] vs 45.6% [28.9%–72.0%]; p<0.0001). This difference was recapitulated by a subsequent randomized controlled study. TACE combined with bicarbonate yielded a 100% objective response rate (ORR), whereas the ORR treated with TACE alone was 44.4% (nonrandomized) and 63.6% (randomized). The survival data suggested that bicarbonate may bring survival benefit.

**Conclusions:** Bicarbonate markedly enhances the anticancer activity of TACE.

**Funding:** Funded by National Natural Science Foundation of China.

**Clinical trial number:** ChiCTR-IOR-14005319.

*For correspondence:
huxun@zju.edu.cn

## Introduction

We recently found that Lactic acidosis could effectively protect cancer cells against glucose starvation or deprivaiton (*Wu et al., 2012*; *Xie et al., 2014*). First, lactic acidosis dramatically reduces glycolysis rate with little wasting glucose to lactate, as such, a limited amount of glucose could support cancer cells for a relatively long time otherwise would be exhausted quickly. Second, when glucose was deprived, lactic acidosis transformed cancer cells to a 'dormant' state, via arresting cells at G0/G1 phase, initiating autophagy, inhibiting apoptosis, etc. The protective function relies on co-presence of lactate and proton, depriving either of which would abolish the function (*Wu et al., 2012*; *Xie et al., 2014*). When converting lactic acidosis to lactosis by a base, the protective function is gone; similarly, removing lactate, acidosis conferred cancer cells with little resistance to glucose deprivation.

The significance of intratumoral lactic acidosis in tumor biology has been extensively revealed by many other investigators. Clinical studies showed that high level of lactate was a strong prognostic

**eLife digest** Surgery is the main treatment for liver cancer, but the most common liver cancer – called hepatocellular carcinoma – can sometimes become too large to remove safely. An alternative option to kill the tumor is to block its blood supply via a process called embolization. This procedure deprives the tumor cells of oxygen and nutrients such as glucose. However, embolization also prevents a chemical called lactic acid – which is commonly found around tumors – from being removed. Lactic acid actually helps to protect cancer cells and also aids the growth of new blood vessels, and so the "trapped" lactic acid may reduce the anticancer activity of embolization.

Previous works suggested that neutralizing the acidic environment in a tumor while depriving it of glucose via embolization could become a new treatment option for cancer patients. Chao et al. now report a small clinical trial that tested this idea and involved patients with large hepatocellular carcinomas. First, a group of thirty patients received the embolization treatment together with an injection of bicarbonate – a basic compound used to neutralize the lactic acid – that was delivered directly to the tumor. The neutralization killed these large tumors more effectively than what is typically seen in patients who just undergo embolization

Chao et al. then recruited another twenty patients and randomly assigned them to receive either just the embolization or the embolization with bicarbonate treatment. This randomized trial showed that the tumors died more and patients survived for longer if they received the bicarbonate together with the embolization treatment compared to those patients that were only embolized. In fact, four patients initially assigned to, and treated in, the embolization-only group subsequently asked to cross over to, and indeed received, the bicarbonate treatment as well.

These data indicate that this bicarbonate therapy may indeed be effective for patients with large tumors that are not amenable to surgery. In future, larger clinical trials will need to be carried out to verify these initial findings.

indicator of increased metastasis and poor overall survival (*Gatenby and Gillies, 2004*; *Brizel et al., 2001*; *Walenta et al., 2000*; *Schwickert et al., 1995*; *Walenta et al., 1997*; *Yokota et al., 2007*; *Paschen et al., 1987*). The work of Gillies and Gatenby group demonstrated that systematic and tumor pHe alkalization could inhibit carcinogenesis, tumor invasion and metastasis, and they also provided integrated models that can predict the safety and efficacy of buffer therapy to raise tumour pHe (*Silva et al., 2009*; *Robey et al., 2009*; *Ibrahim-Hashim et al., 2012*) and related theoretical work (*Martin et al., 2012*, *Martin et al., 2011*). Furthermore, many studies reported that lactic acidosis played multifaceted roles in skewing macrophages (*Colegio et al., 2014*) and inhibiting the function of cytotoxic T cells (*Haas et al., 2015*), altering cancer cell metabolism (*Chen et al., 2008*; *Sonveaux et al., 2008*), inducing chromosomal instability (*Dai et al., 2013*), and promoting tumor angiogenesis (*Gatenby and Gillies, 2004*; *Végran et al., 2011*).

According to the guideline of Barcelona Clinic Liver Cancer (BCLC) staging and treatment strategy, HCC larger than 3 cm in diameter is not suitable for curative therapy (surgical resection, liver transplantation, and ablation) and the recommended treatment is TACE (*Forner et al., 2012*; *El-Serag, 2011*; *Knox et al., 2015*). But sadly, it is recognized that TACE is not effective to treat large tumors (*Sieghart et al., 2015*). This leaves the patients with large HCC without choice of effective therapy, as also pointed out by Sieghart et al, "maximal restriction of patients selection for TACE would otherwise only improve the results of the treatment modality per se but again would leave those more advanced patients within the intermediate stage without treatment options." (*Sieghart et al., 2015*)

TACE is for local control of the targeted tumor. TEX *equation check*TACE kills HCC via 2 mechanisms, delivering concentrated anticancer drugs locally into tumor and occluding tumor feeding arteries to deprive nutrients to starve cancer cells. We would focus on the second mechanism. Occluding tumor feeding arteries effectively deprive nutrients including glucose. The problem is that embolization-created hypoxia condition would stimulate cancer cells to emit strong signals to initiate angiogenesis (*Knox et al., 2015*) to reestablish tumor vasculature to bypass the occluded tumor feeding arteries. If the tumor cells cannot be rapidly eliminated, tumor vasculature would be reestablished, and a certain amount of tumor (ranging from a few percent of the original tumor to

**Table 1.** Clinical and tumor characteristics of patients treated with cTACE and TILA-TACE in the nonrandomized study.

| Variables | Patients | |
|---|---|---|
| | TILA-TACE | cTACE |
| Patient number | 30 | 27 |
| Median age, years | 57 (Range 32–81) | 54 (Range 37–81) |
| Gender (M/F) | 27/3 (90.0%/10.0%) | 27/0 (100%/0%) |
| Aetiology | | |
| HBV | 24 (80.0%) | 25 (92.6%) |
| HCV | 0 (0%) | 0 (0%) |
| Non B-non C | 6 (20.0%) | 2 (7.4%) |
| Cirrhosis (radiology) | 30 (100%) | 27 (100%) |
| Bilirubin, μM | 16.9 ± 9.4 | 22.5 ± 11.6 |
| Albumin, g/L | 39.0 ± 6.9 | 37.4 ± 5.3 |
| AST, U/L | 74.9 ± 102.3 | 83.5 ± 54.1 |
| ALT, U/L | 54.1 ± 80.4 | 67.3 ± 43.5 |
| AFP, >400 ng/mL | 9 (30.0%) | 15 (55.6%) |
| Child-Pugh class, A/B | 27/3 (90.0%/10.0%) | 25/2 (92.6%/7.4%) |
| The size of largest tumor (cm) | 9.2 (range 5.0–13.6) | 10.3 (range 5.0–14.6) |
| Tumor >10 cm | 14 (46.7%) | 15 (55.6%) |
| Tumor 5~10 cm | 16 (53.3%) | 12 (44.4%) |
| Multifocal tumors in 1 lobe | 8 (26.7%) | 12 (44.4%) |
| Multifocal tumors in 2 lobes | 8 (26.7%) | 12 (44.4%) |
| BCLC stage | | |
| B | 19 (63.3%) | 18 (66.7%) |
| C | 11 (36.7%) | 9 (33.3%) |
| Macrovascular invasion | 5 (16.7%) | 4 (14.8%) |
| The right branch of portal vein | 4 (13.3%) | 2 (7.4%) |
| Hepatic vein | – | 1 (3.7%) |
| The right branch of portal + hepatic vein | 1 (3.3%) | 1 (3.7%) |
| Extra-hepatic metastasis | 8 (26.7%) | 8 (29.6%) |
| Lung | 1 (3.3%) | 6 (22.2%) |
| Lung + bone | 1 (3.3%) | 0 (0%) |
| Soft tissue | – | 0 (0%) |
| Lymph nodes | 5 (12.2%) | 0 (0%) |
| Bone | 1 (2.0%) | 1 (3.7%) |
| Bone+lymph node | – | 1 (3.7%) |

HBV, hepatitis B virius;
HCV, hepatitis C virius;
AST, Aspartate transaminase;
ALT, Alanine aminotransferase;
AFP, alpha-feto-protein.

even a larger tumor known as progressive disease) would survive and thrive. The lactate concentrations in HCC biopsies were around 20 mM (unpublished data), suggesting a lactic acidosis condition. After TACE, lactic acidosis would be trapped in embolized tumor and it would potentially attenuate the therapeutic efficacy of TACE. If it were true, locally infusion of bicarbonate to neutralize it would result in a severer necrosis in the embolized area.

**Table 2.** Geometric means of viable tumor residues after treatment of cTACE or TILA-TACE in the nonrandomized cohort of 57 patients.

|  | Geometric mean (95% CI) | | |
|---|---|---|---|
|  | cTACE (n=27) | TILA-TACE (n=30) | p value |
| Crude VTR | 45.1% (30.3%–67.0%) | 7.1% (4.4%–11.5%) | <0.0001 |
| Multivariable adjusted VTR* | 45.6% (28.9%–72.0%) | 7.1% (4.6%–10.9%) | <0.0001 |

VTR: viable tumor residues;

cTACE: transarterial chemoembolization;

TILA-TACE: targeting-intratumoral-lactic-acidosis TACE;

*Adjusted for viable tumor volume prior to treatment and macrovascular invasion using the general linear regression. No appreciable alterations in results were found after adjustment for other covariates such as age, BCLC tumor stage, extra-hepatic metastasis, HBV DNA copy numbers, and tumor multifocality.

## Results

### Local control of targeted large tumors as assessed by viable tumor residues (VTR)

TACE is to control the targeted tumors rather than to systematically control the disease. As such, to measure the necrosis of the targeted tumors can quantitatively reflect the therapeutic efficacy of TACE. It is known that large size of HCC is a major obstacle to limit the therapeutic efficacy of TACE (*Sieghart et al., 2015*), the larger the tumor size, the poorer the efficacy of TACE. Thus, the large HCC is a perfect tumor model to test our hypothesis. If intratumoral lactic acidosis is responsible for the therapeutic limitation, locally neutralizing it should significantly improve the anticancer activity, which can be quantitatively measured by the viable tumor residues. To the best of our knowledge, assessment of the necrosis of tumor after the first treatment is probably most objective with least interferences of known and unknown factors. Unless otherwise indicated, the data presented below are the response to the first TACE treatment.

We retrieved 27 patients with large HCC (ranging 5.0–14.6 cm) from the pool of patients treated with cTACE in the hospital between 2010 and 2012 according to the inclusion and exclusion criteria as specified in the Materials and methods section. The amount of viable tumor residues after cTACE treatment was calculated. We then recruited 30 patients for the TILA-TACE group using the same inclusion and exclusion criteria between 2012 and 2013. Most of patients' demographic and clinico-pathologic characteristics between the two treatment groups cTACE and TILA-TACE were generally comparable (Table 1). However, patients in the cTACE group were more likely to have multifocal tumors.

The geometric mean of VTR after the first treatment was 45.1% (95% CI: 30.3%–67.0%) in the cTACE group, significantly greater than that in the TILA-TACE (7.1% [95% CI: 4.4%–11.5%]; p<0.0001) (*Table 2* and *Figure 1*). We further evaluated whether treatment effects (VTR) were confounded by some covariates such as age, BCLC tumor stage, extra-hepatic metastasis, HBV DNA copy numbers, viable tumor volume before treatment, macrovascular invasion, and tumor multifocality using general linear models. In this study, none of these clinical covariates were significantly associated with VTR (data not shown). Adjustment of these variables did not appreciably alter the results (*Table 2*). We then calculated the relative therapeutic improvement by TILA-TACE as described in Materials and methods. TILA-TACE achieved a 81.1% therapeutic improvement relative to cTACE.

We also categorized tumor responses to treatment into four categories from complete response (CR) to progressive disease (PD) according to EASL criteria. Complete or partial responses to treatment in the TILA-TACE group were significantly higher than that in the cTACE group. The percentage of CR, PR, SD, and PD in the cTACE group was 0%, 44.4%, 33.3%, and 22.2% (*Figure 2A*), respectively, compared with 23%, 77%, 0%, and 0%, respectively, in the TILA-TACE group (*Figure 2B*). The total objective response rate (complete or partial responses) in the cTACE group was 44.4%, whereas it was 100% in the TILA-TACE group (*Figure 2C*). The observed greater responses to treatment in the TILA-TACE than in the cTACE persisted after accounting for tumor volume prior to treatment using the proportional odds model (p<0.0001).

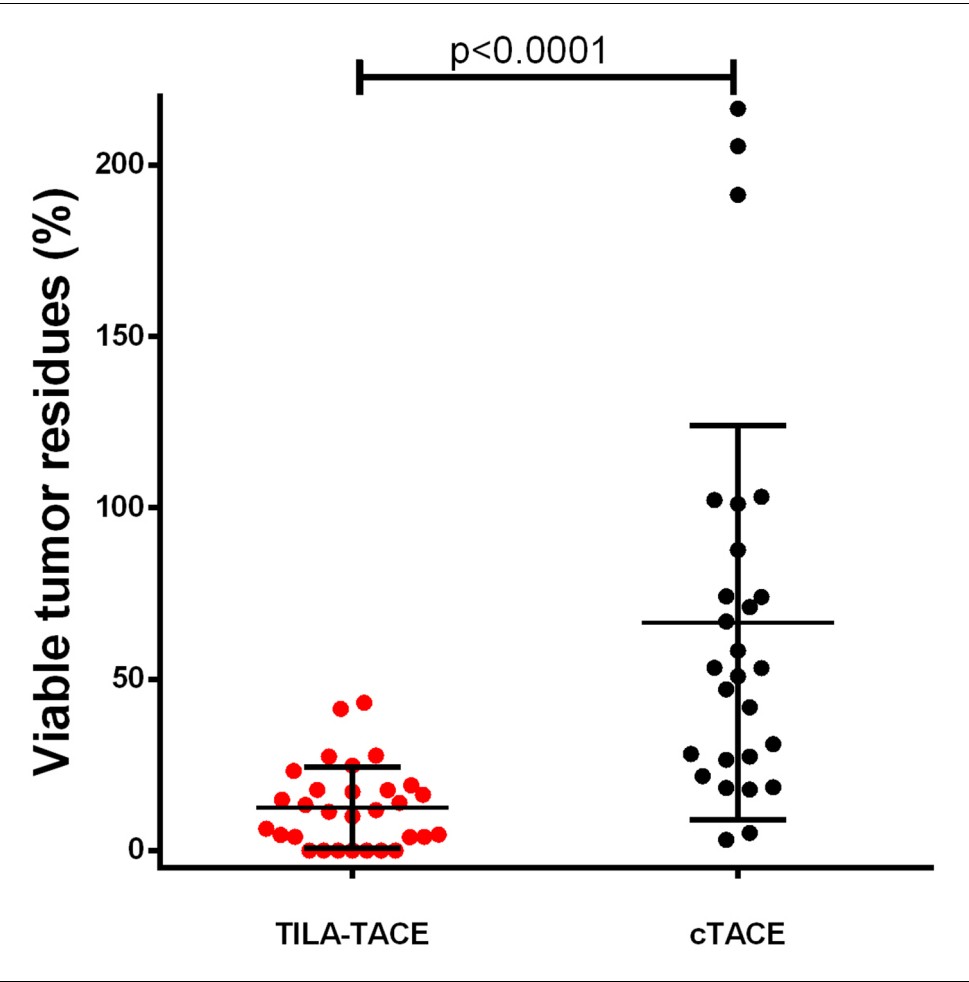

**Figure 1.** Viable tumor residues (VTR) after treatment of cTACE verus TILA-TACE. Patients' demographic parameters were described in *Table 1*. The relative therapeutic improvement by TILA-TACE was 81.1%. Relative therapeutic improvement by bicarbonate is defined as $[(\mu_1-\mu_2)/\mu_1] \times (100\%)$, where $\mu_1$ is the mean of viable tumor residues after treatment (cTACE) and $\mu_2$ the mean residual after treatment (TILA-TACE), and the maximal therapeutic improvement is 100%. Differences in VTR between two groups were statistically significant (p<0.0001), as assessed using the general linear model after adjustment for viable tumor volume before treatment

The online version of this article includes the following source data for figure 1:

**Source data 1.** Calculation of viable volume residues of each patient in the nonrandomized study after the first round treatment.

Because of the marked therapeutic improvement by TILA-TACE, the sample size required for the subsequent RCT to evaluate tumor responses to treatment was rather small. When patient number was 10 for each group, the estimated power value reached 0.826. In 2014, we recruited and randomly assigned twenty patients with large HCC (5.0 – 13.5 cm) into cTACE (n=10) or TILA-TACE (n=10). Again, most of the selected clinicopathologic characteristics were well matched between two treatment groups (*Table 3*). After completion of the first treatment, the VTR in the TILA-TACE group was 5.5-fold lower than that in the cTACE group (4.6% [95% CI: 1.8%–11.4%] vs. 25.4% [95% CI: 10.1%–64.0%]; p=0.008) (*Figure 3* and Table 4). We also evaluated whether the treatment effect was confounded by aforementioned clinical parameters in the RCT. Among them, only viable tumor volume before treatment was significantly associated with VTR. Adjustment for tumor volume before treatment slightly changed the results, with the VTR in the TILA-TACE of 4.1% (2.0%–8.4%) vs. 28.1% (13.9%–56.8%) in the cTACE (p=0.0009). The relative therapeutic improvement by bicarbonate in the RCT was 80.1%, similar to that observed in the non-randomized cohort, 81.1%.

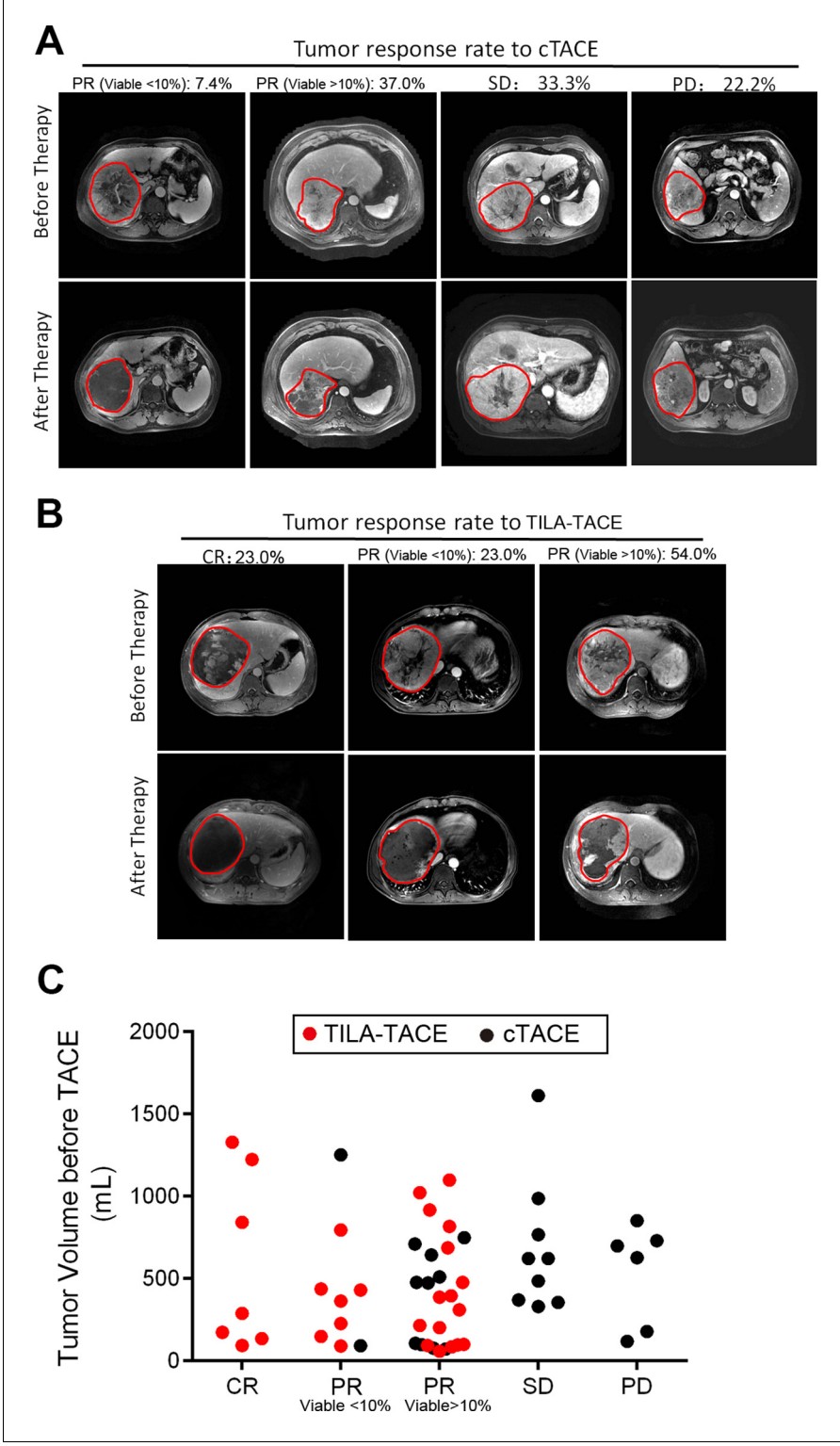

**Figure 2.** Tumor objective response to cTACE or TILA-TACE categorized according to EASL criteria. Rate of tumor response to treatment and representative MRI images of tumor before and after treatment (**A**) in the cTACE group (n=27) and (**B**) in the TILA-TACE group (n=30). (**C**) The difference in tumor responses between two groups was statistically significant (p<0.0001), as assessed by the proportional odds model after adjustment for viable tumor volume before treatment. CR, complete necrosis; PR, viable tumor volume less than 50% of the viable tumor
*Figure 2 continued on next page*

*Figure 2 continued*

volume before treatment; SD, viable tumor volume larger than 50% of the viable tumor volume before treatment; PD, viable tumor volume larger than 100% after treatment.

The online version of this article includes the following source data for figure 2:

**Source data 1.** The criteria and classification of response to treatment in the nonrandomized study after the first round treatment.

If assessed by tumor objective response rate according to the EASL criteria, in TILA-TACE group, out of 12 targeted tumors, 4 achieved CR and 8 PR (*Figure 4*), whereas in cTACE group, out of 11 targeted tumors, 1 achieved CR, 6 PR, 2 SD, and 2 PD (*Figure 4*), with *P* value of 0.003 after

**Table 3.**

| Variables | Patients | |
|---|---|---|
| | TILA-TACE | cTACE |
| Patient number | 10 | 10 |
| Median age, years | 58 (Range 40–86) | 53 (43–81) |
| Gender (M/F) | 9 /1 (90.0%/10.0%) | 7 /3 (70.0%/ 30.0%) |
| Aetiology | | |
| HBV | 9 (90.0%) | 8 (80.0%) |
| HCV | 0 (0%) | 0 (0%) |
| Non B-non C | 1 (10.0%) | 2 (20.0%) |
| Cirrhosis (radiology) | 10 (100%) | 10 (100%) |
| Bilirubin, μM | 17.2 ± 10.1 | 16.7 ± 7.6 |
| Albumin, g/L | 38.7 ± 3.1 | 38.1 ± 5.2 |
| AST, U/L | 64.8 ± 44.8 | 52.8 ± 19.2 |
| ALT, U/L | 60.6 ± 48.0 | 41.0 ± 29.9 |
| AFP, >400 ng/mL | 3 (30.0%) | 4 (40.0%) |
| Child-Pugh class, A/B | 10/0 (100%/0%) | 10/0 (100%/0%) |
| The size of largest tumor (cm) | 7.9 (range 5.0–13.5) | 7.5 (range 5.0–13.0) |
| Tumor >10 cm | 3 (30.0%) | 3 (30.0%) |
| Tumor 5~10 cm | 7 (70.0%) | 7 (70.0%) |
| Multifocal tumors in 1 lobe | 3 (30.0%) | 3 (30.0%) |
| Multifocal tumors in 2 lobes | 4 (40.0%) | 2 (20.0%) |
| BCLC stage | | |
| B | 7 (70.0%) | 8 (70.0%) |
| C | 3 (30.0%) | 2 (20.0%) |
| Macrovascular invasion | 2 (20.0%) | 1 (10.0%) |
| The right branch of portal vein | 1 (10.0%) | 1 (10.0%) |
| The left branch of portal vein | 1 (10.0%) | 0 (0%) |
| Extra-hepatic metastasis | 1 (10.0%) | 2 (20.0%) |
| Brain | 0 (0%) | 1 (10.0%) |
| Lymph nodes | 1 (10.0%) | 1 (10.0%) |

HBV, hepatitis B virius;
HCV, hepatitis C virius;
AST, Aspartate transaminase;
ALT, Alanine aminotransferase;
AFP, alpha-feto-protein.

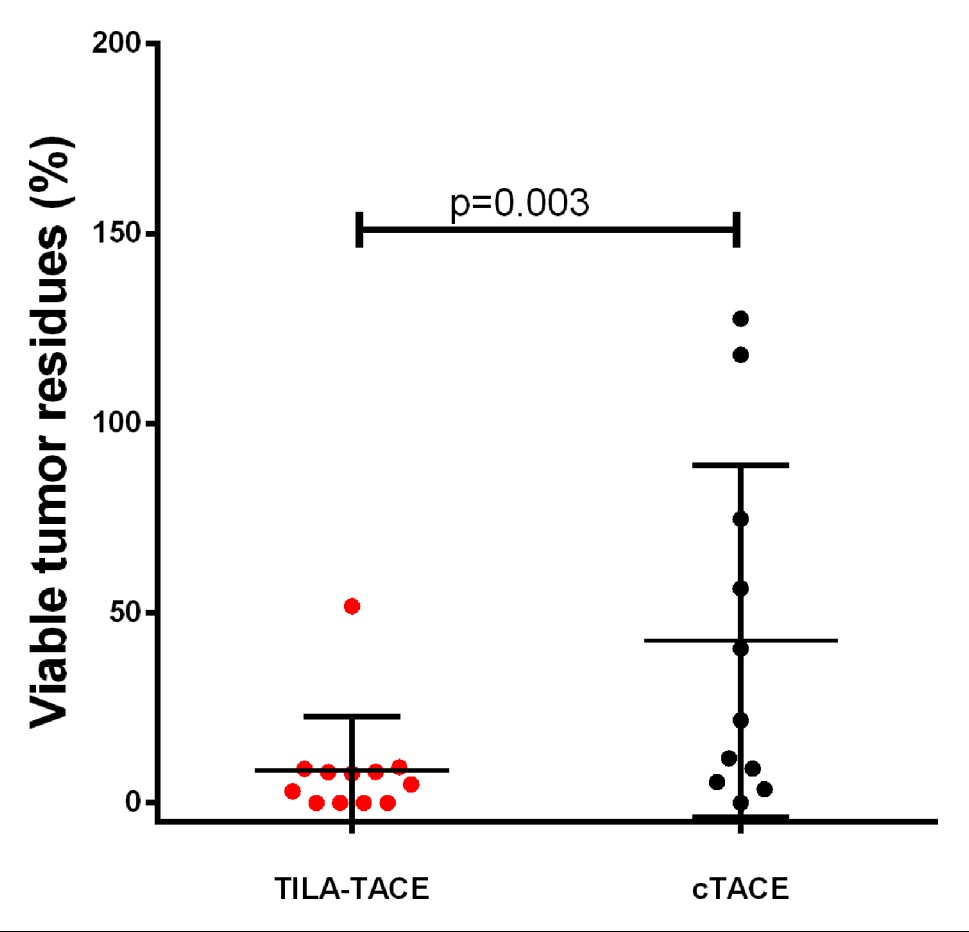

**Figure 3.** Viable tumor residues after treatment with cTACE or TILA-TACE in the randomized controlled study. The relative therapeutic improvement by TILA-TACE was 80.1%. Differences in VTR between two arms were statistically significant (p=0.0009), as assessed using the general linear model after adjustment for viable tumor volume before treatment.

The online version of this article includes the following source data for figure 3:

**Source data 1.** Calculation of viable volume residues of each patient in the randomized study after the first round treatment.

adjustment for tumor volume prior to treatment. A similar result was found when assessing treatment effects for the largest tumor (p=0.003). In this RCT, the ORR in TILA-TACE group was 100% and the ORR in cTACE group was 63.6%.

## Overall survival

In the nonrandomized cohort of study, the 1-, 2-, 3-year survival of 27 patients treated with cTACE retrieved from our database were 66.7% (95% CI 45.7%–81.1%), 40.7% (95% CI 22.5%–58.2%), and 25.9% (95% CI 11.5%–43.1%), respectively, with a median survival of 14 months (*Figure 5A*), and the 1-, 2-, 3-year survival of 30 patients treated with TILA-TACE were 82.8% (95% CI 63.4%–92.8%), 67.7% (95% CI 47.0%–81.8%), and 61.8% (95% CI 39.7%–77.8%), respectively, with a median survival beyond 41 months (*Figure 5A*). The survival difference between TILA-TACE and cTACE was statistically significant (p=0.0052).

In the randomized study, 4 patients initially assigned in and treated with cTACE group subsequently requested to cross over to TILA-TACE treatment. Although the cross-over was ethically warranted, this somehow blurred the overall survival difference between cTACE and TILA-TACE (*Figure 5B*). In cTACE group, 3 deaths occurred in 6 patients who solely received cTACE treatment, and 4 patients who initially received cTACE treatment and subsequently crossed over to TILA-TACE

**Table 4.** Geometric means of viable tumor residues after treatment of cTACE or TILA-TACE in the RCT.

| | Geometric mean (95% CI) | | |
| --- | --- | --- | --- |
| | cTACE (n=10) | TILA-TACE (n=10) | p value |
| Crude VTR | 25.4% (10.1%–64.0%) | 4.6 (1.8%–11.4%) | 0.008 |
| Multivariable adjusted VTR* | 28.1% (13.9%–56.8%) | 4.1 (2.0%–8.4%) | 0.0009 |

VTR: viable tumor residues;

cTACE: transarterial chemoembolization;

TILA-TACE: targeting-intratumoral-lactic-acidosis TACE;

RCT: randomized clinical trial.

*Adjusted for viable tumor volume prior to treatment using the general linear regression. No appreciable alterations in results were found after adjusting for other covariates such as age, BCLC tumor stage, extra-hepatic metastasis, HBV DNA copy numbers, macrovascular invasion, and tumor multifocality individually.

treatment were alive. In TILA-TACE group of 10 patients, 3 deaths occurred and 7 patients live. There was no apparent difference in overall survival between two treatment groups in the intent-to-treat (ITT) analysis (*Figure 5B*, left panel). However, a survival advantage appears in TILA-TACE treatment over cTACE treatment in the per-protocol (PP) analysis (*Figure 5B*, right panel), but statistically not significant. Overall, the RCT was limited by the small sample size and the result of the PP analysis was potentially confounded by the crossover of patients from cTACE group to TILA-TACE.

After pooling all patients together, there was a significant difference of survival between cTACE and TILA-TACE group (*Figure 5C*).

## Adverse effects

The adverse effect between cTACE and TILA-TACE group were comparable (*Tables 5* and *6*).

## Discussion

In this clinical investigation, we carried out 2 studies sequentially, the first one is a nonrandomized controlled study, which demonstrated a remarkable therapeutic improvement of TILA-TACE, based on which, a randomized controlled study was designed and carried out, and again it demonstrated a superior anticancer activity of TILA-TACE. The most striking point is that the numbers reflecting the therapeutic improvements by TILA-TACE from the 2 studies were nearly identical (81.1% and 80.1%). This confirms the consistency of anticancer activities of cTACE as well as TILA-TACE with respect to local control of large HCC.

We compared the ORR in our cTACE practice with those reported globally. The average objective tumor response to TACE is 35% (range, 16%–61%), as systematically reviewed by Llovet and Bruix for the Barcelona-Clinic Liver Cancer Group in 2002 (*Llovet and Bruix, 2003*). In 2012, Forner, Llovet, and Bruix summarized that more than 50% of patients had an objective response to TACE (*Forner et al., 2012*), suggesting that the objective tumor response to TACE in the 10-year period (2002–2012) had been increased for about 15%. The complete tumor response to TACE is rare (0-4.8%) (*Jansen et al., 2005*). Obviously, the results obtained from our cTACE practice were similar to those reported globally.

TACE is for local control of the targeted tumor. Many previous studies have confirmed that better local control was an independent prognostic indicator for patient survival (*Kim et al., 2015*; *Jung et al., 2013*; *Kim et al., 2013*; *Shim et al., 2012*; *Riaz et al., 2011*; *Gillmore et al., 2011*; *Riaz et al., 2010*). Kim et al and Shim et al (*Kim et al., 2015*; *Shim et al., 2012*) further demonstrated a clear prognostic difference between CR, PR, stable disease and progressive disease. The current study demonstrated that TILA-TACE achieved a remarkable improvement of local tumor control and suggested an early sign of improved survival for patients with large HCCs (*Figure 5A*). It is noted that, during follow up, 16 patients in the TILA-TACE arm (*Figure 5A*), eventually exhibited progressive disease, including 11 patients with new foci in the liver, 1 with new liver foci and lung metastasis, and 3 with lung metastasis, and 1 with bone metastasis, all of which may account for the death (*Supplementary file 2*). These observations suggest that fast CR and timely control of

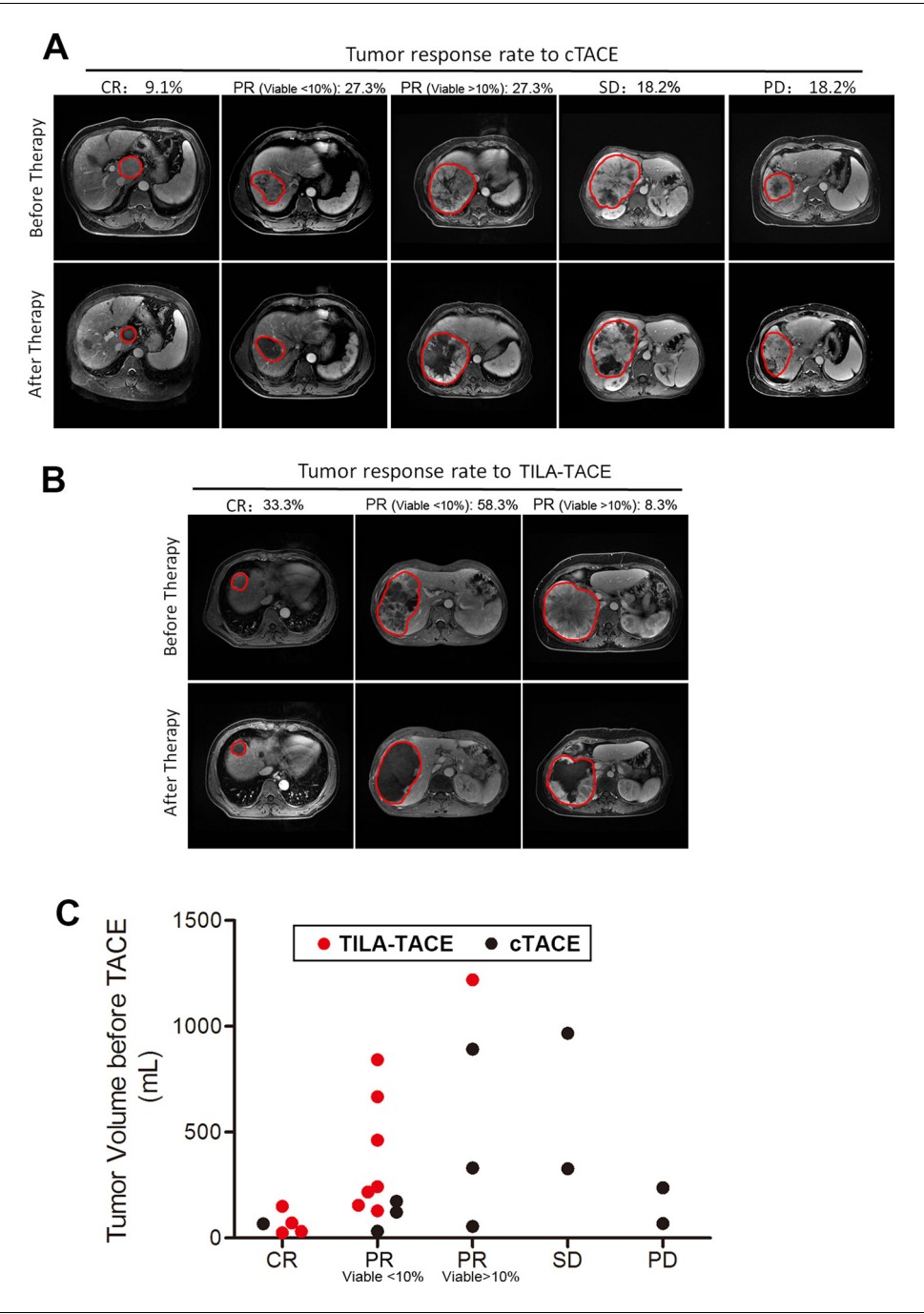

**Figure 4.** Tumor objective response to cTACE or TILA-TACE according to EASL criteria. Twenty patients were randomly assigned to treatment of cTACE or TILA-TACE. (**A**) Tumor response rate to cTACE and representative MRI images of tumor before and after treatment. 10 patients were treated with cTACE. (**B**) Tumor response rate to TILA-TACE and representative MRI image of tumor before and after treatment. 10 patients were treated with TILA-TACE. (**C**) The pattern of tumor response to TILA-TACE and cTACE. The difference between 2 groups was statistically significant (p=0.003), as assessed using proportional odds model. CR, complete necrosis; PR, viable tumor volume less than 50% of the viable tumor volume before treatment; SD, viable tumor volume larger than 50% of the viable tumor volume before treatment; PD, viable tumor volume larger than 100% after treatment. The online version of this article includes the following source data for figure 4:

**Source data 1.** The criteria and classification of response to treatment in the randomized study after the first round treatment.

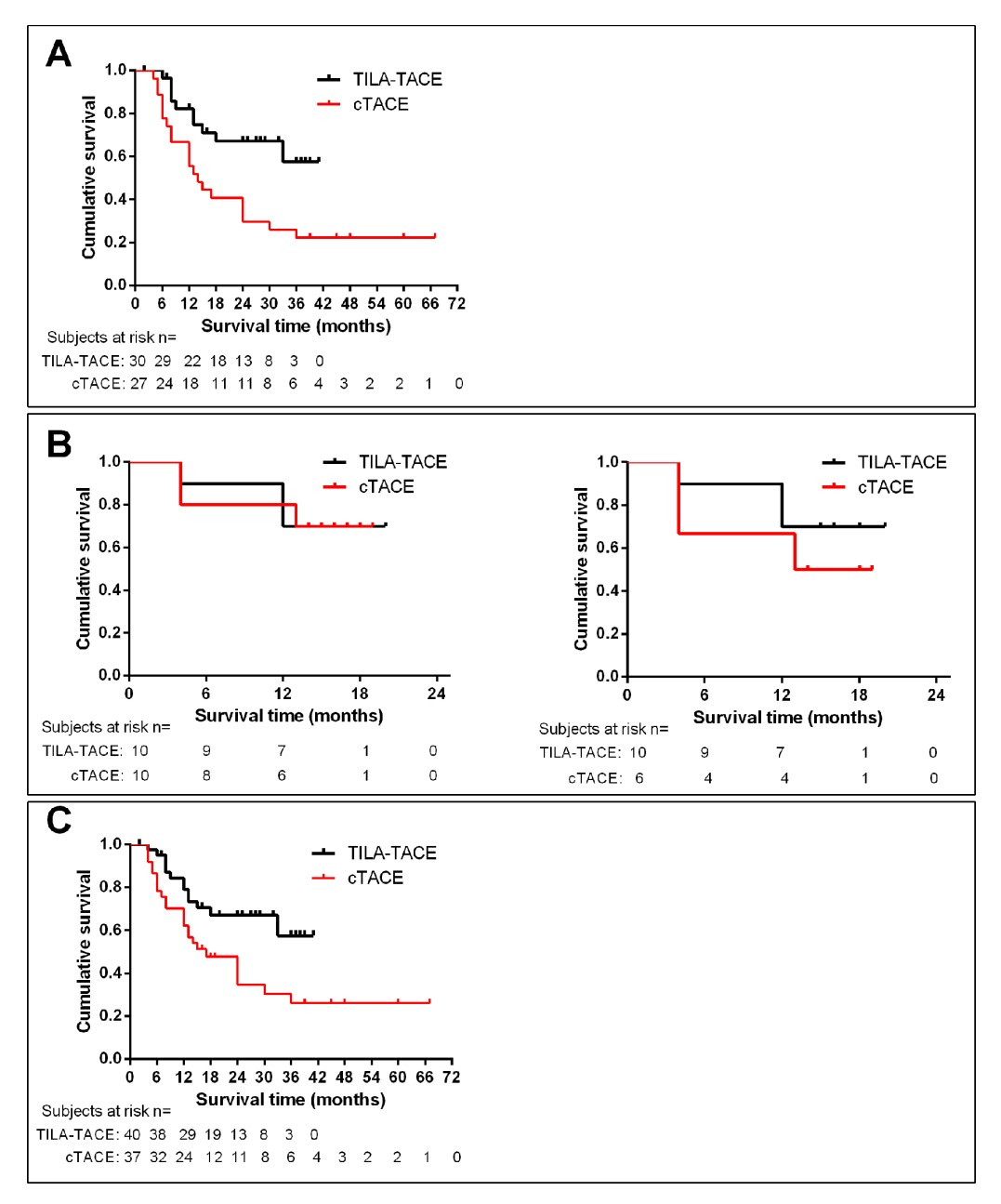

**Figure 5.** Kaplan-meier analysis of cumulative survival of patients receiving TILA-TACE or cTACE treatment. (**A**) Cumulative survival of patients described in *Table 1*. p=0.0052. (**B**) Survival of patients described in *Table 2*. Left panel, all patients; right panel, patients who initially assigned to cTACE but subsequently crossed over to TILA-TACE treatment were excluded. p>0.05. (**C**) Cumulative Survival of patients pooled from *Table 1* and *2*. p=0.0133.

The online version of this article includes the following source data for figure 5:

**Source data 1.** The survival status of each patient at the cut-off date in the nonrandomized and the randomized studies.

recurrent tumors in the liver would likely improve the survival of patients, especially those with large tumor burden and low liver reserve.

Nevertheless, the work is limited by the study design. The randomized controlled study designed in this investigation was for local tumor control, not for survival. Although the small sample size allowed us to evaluate the local tumor control, we did not expect that such small sample size would yield statistically significant data for cumulative survival. Evaluation of survival is a matter much more complicated than evaluation of local control. There are many more factors affecting the survival of

**Table 5.** Adverse events of patients receiving TILA-TACE or cTACE.

| Adverse events* | TILA-TACE | cTACE[†] |
|---|---|---|
| Pain | 5 out of 30 | 3 out of 27 |
| Fever ($\geq$38.5)[‡] | 13 out of 30 | 9 out of 27 |

* The adverse events monitored also include acute hepatic decomposition, irreversible hepatic decompensation, respiratory failure or decompensation, biliary stricture or obstruction, liver abscess, gastrointestinal bleeding, arterial thrombosis, arterial-portal shunting. These events were not observed in the patients.

[†] Patients were retrospectively retrieved from our database.

[‡] Mild fever occurred to all patients.

patients than those affecting the local tumor control, e.g., tumor characteristics, liver function reserve, tumor staging, disease complications, vascular invasion, metastasis, etc., all of which must be well controlled. While we acknowledged these limitations of the present small RCT, the preliminary survival data allowed us to rationally calculate the sample size for a subsequent large RCT for evaluating the survival difference between cTACE and TILA-TACE. We are planning a large-scale RCT to further confirm the therapeutic advantage of TILA-TACE with respect to overall and progression-free survival of patients with large HCCs.

There was no significant difference of adverse effect between TILA-TACE and cTACE (*Tables 3* and *4*), i.e., TILA-TACE was as tolerable as cATCE. It was within our expectation, as locally administration of bicarbonate into tumor is safe.

Taken together, this pilot study demonstrated that bicarbonate infusion locally into HCC can markedly enhance anticancer activity of TACE, supporting the notion that neutralizing intratumoral lactic acidosis combined with glucose deprivation may deliver an effective approach to control tumor.

# Materials and methods

## Patients and study design

The study was performed with patients' written informed consent and with the approval of hospital's Institutional Review Board. Patients' consent to publish is obtained. The study is composed of 2 parts (Reporting standard 1). In the first part, twenty seven patients treated with cTACE were retrospectively retrieved (February 2010 – March 2012) (*Supplementary file 1*) and the viable tumor residues were calculated, and 30 patients (January 2012 – September 2013) (*Supplementary file 2*) with large HCC were recruited and treated with TILA-TACE (targeting-intratumoral-lactic-acidosis TACE, in this modality, bicarbonate was infused into tumor to neutralizing intratumoral lactic acidosis). In the second part, the data generated from part 1 were used to estimate the sample size for a subsequent randomized controlled study. Twenty patients (March 2014 – August 2014) (*Supplementary file 3*) were randomly assigned to either TILA-TACE or cTACE treatment with ratio 1:1 (registration number: ChiCTR-IOR-14005319 in Chinense Clinical Trial Registry; protocol [*Supplementary file 5*] can be accessed at http://www.medresman.org/), according to the method of random number table. Patient inclusion criteria are: a diagnosis of HCC based on EASL or histological evidence, *Barcelona* Clinic Liver Cancer (BCLC) stage B or C, Child-pugh A or B, adult patients of age $\geq$ 20, hypervascular lesion as evaluated by triphasic MRI and *digital* subtraction

**Table 6.** Adverse events in patients in the RCT.

| Adverse events* | TILA-TACE | cTACE |
|---|---|---|
| Pain | 2 out of 10 | 1 out of 10 |
| Fever ($\geq$38.5)[†] | 2 out of 10 | 3 out of 10 |

* The adverse events monitored also include acute hepatic decomposition, irreversible hepatic decompensation, respiratory failure or decompensation, biliary stricture or obstruction, liver abscess, gastrointestinal bleeding, arterial thrombosis, arterial-portal shunting. These events were not observed in the patients.

[†] Mild fever occurred to all patients.

angiography (DSA). Patient exclusion criteria are: BCLC stage 0, A or D, Child pugh C, evidence of combined A-V shunt.

## Sample size calculation in the RCT

Sample size calculation was determined by the values of power and alpha: power (0.8) and alpha (0.05) can properly tell the statistical significance and minimize the number of patients to receive false choice of therapy. In order to calculate sample size for RCT, we calculate the viable tumor residues of 30 patients treated with TILA-TACE and 27 patients treated with cTACE as described below (Evaluation of tumor response, Calculation of total tumor volume, viable tumor volume, and necrotic tumor volume). Based on these data, we assessed the minimal number of patients required in a subsequent RCT. We assume the probability of making a Type I error to be 0.05 (α, two-sided), and the probability of making a Type II error to be 0.20 (β level), so the power of this RCT is 0.8 (1-β). The sample size was calculated according to Schouten's general formula:

$$n_1 \geq \left(z_{1-\frac{\alpha}{2}} + z_\beta\right)^2 \frac{(\tau + \gamma)\sigma_1^2}{\gamma(\mu_1 - \mu_2)^2} + \frac{(\tau^2 + \gamma^3)z_{1-\alpha/2}^2}{2\gamma(\tau + \gamma)^2}$$

Where $n_1$ and $n_2$ represent the number of patients assigned to cTACE and TILA-TACE group, $\gamma = n_1/n_2$, $\mu_1$ (cTACE) and $\mu_2$ (TILA-TACE) is the mean viable tumor residues, and $\sigma_1$ and $\sigma_2$ are the corresponding standard deviation, $\tau = \sigma_2^2/\sigma_1^2$, and Z is the normal deviate for alternative hypothesis at a level of significance. In practice, we used PASS software (version 11) to calculate the power and sample size, using the parameters mentioned above. As can been seen in *Supplementary file 4*, 9 patients in each group already satisfy the power and alpha value. For a pilot study, the sample size of 10 patients for each group was appropriate, as the estimated power value reached 0.826. Twenty patients were randomly allocated in a ratio of 1:1 to each group for the RCT.

## Primary endpoint and secondary endpoint

As the study was designed for observing the local tumor control, the primary endpoint was tumor objective response rate to the first treatment, measured by viable tumor residue (VTR), and the secondary endpoint was the overall survival.

## Definition of relative therapeutic improvement

Relative therapeutic improvement by bicarbonate is defined as $[(\mu_1 - \mu_2)/\mu_1] \times (100\%)$, where $\mu_1$ is the mean percentage of viable tumor residues after treatment (cTACE) and $\mu_2$ the mean percentage of viable tumor residues after treatment (TILA-TACE). The maximum therapeutic improvement is 100%.

## Quantitative calculation of viable tumor residues after treatment

The enhanced area and nonenhanced area in every slice of a tumor (2 mm thichness of each image if scanned by 3.0T MRI Discovery 750, GE Medical Systems or 3.5 mm thichness of each image if scanned by 3.0T MRI Signa Excite HD,GE Medical Systems), which was confirmed by 2 radiologists, was integrated according to MIPAV software (*Partecke et al., 2011*) (http://mipav.cit.nih.gov/). Summation of integrated enhanced and nonenhanced area of all slices of a tumor gave the viable and necrotic volumes of a tumor. Total tumor volume was the sum of viable and necrotic volume.

Viable tumors were assessed by MRI according to EASL criteria. The enhanced and non-enhanced areas represent viable and necrotic tumors. MRI examinations were performed on a 3.0T MRI (Signa Excite HD,GE Medical Systems,USA) or 3.0T MRI (Discovery 750, GE Medical Systems, USA). We assessed the necrotic and non-necrotic volume using T1 post gadolinium. In some cases, when the T1 post gadolinium image of the lesions did not show typical enhancement, we confirmed these lesions using imaging sequence of DWI and T2W.

The parameters of 3.0T MRI (Signa Excite HD,GE Medical Systems,USA) were as follows: T1WI fast gradient echo sequence TR/TE 180/2.4, FOV 40 × 36 cm, slice thickness 7 mm; T2WI fast spin echo sequence, TR/TE 6000/104.2, FOV 40 × 28 cm, slice thickness 7 mm; DWI single-shot spin-echo echo-planar imaging, SS-SE-EP, b value 600 sec/mm², TR/TE 1300/52.3, FOV 40 × 40 cm, slice thickness 7 mm; Dynamic contrast-enhanced LAVA sequence, inversion time 5.0 s, TR/TE 2.7/1.3, FOV 40 × 40 cm; slice thickness 4 mm.

The parameters of 3.0T MRI (Discovery 750, GE Medical Systems, USA) were: T1WI fast gradient echo sequence, TR/TE 4.2/1.9, FOV 36 36 cm, slice thickness 4 mm; T2WI fast spin echo sequence, TR/TE 6666/65.3, FOV 36 × 36 cm, slice thickness 5 mm; DWI single-shot spin-echo echo-planar imaging, SS-SE-EP, b value 800 sec/mm$^2$, TR/TE 6000/53.1; FOV 36 × 36 cm, slice thickness 5 mm; Dynamic contrast-enhanced LAVA sequence, inversion time 5.0 s, TR/TE 3.8/1.6, FOV 36 × 36 cm, slice thickness 4 mm.

## European association for the study of the liver (EASL) criteria of tumor response to treatment

The targeted-tumor response to treatment was assessed 30 days after the first treatment according to EASL criteria (Bruix et al., 2001) and defined as below: complete response (CR), no obvious viable residues; partial response (PR), viable residues <50%; stable disease (SD), viable tumor residues between >50% but ≤100%; and progressive disease (PD), viable tumors >100%.

## Overall survival

Overall survival was measured from the date of the first treatment until the date of death or the final follow up visit.

## Treatment

TILA-TACE was performed through the transfemoral route using a 5-Fr catheter (Shepherd-hook modified Angiographic Catheter, HANACO Medical, Tian Jin, China) that was advanced to celiac artery. The tumor feeding arteries were defined by digital subtraction angiography. Then, a coaxial microcatheter (2.8 Fr Marguerite II, ASAHI INTECC GMA CO., LTD, Nagoya, Japan) was selectively inserted through a 5-Fr catheter into the tumor feeding artery, into which, 5% sodium bicarbonate (5% Sodium Bicarbonate Injection, Hunan Kelun Pharmaceutical, Ltd., Hunan, China) was infused alternatively with doxorubicin-lipiodol emulsion and oxaliplatin/homocamptothecin with the dose adjusted to tumor size . The dose of bicarbonate ranged between 50 and 250 ml corresponding to tumor sizes between 5 and 14 cm. Finally, the artery was permanently embolized with PVA of proper sizes (Embosphere, BioSphere Medical, Paris, France) and microcoil (Tornado, COOK Medical, USA).

The following example may give a clearer description of the procedure:

If a tumor (10 cm) is to be treated, according to our experience, we would prepare 150 ml 5% bicarbonate, 60 ml lipiodol doxorubicin emulsion (40 mg doxorubicin dissolved in 30 ml contrast medium and mixed with 30 ml lipiodol), oxaliplatin 150 mg in 20 ml 5% glucose, 40 mg homocamptothecin in 20 ml saline, PVA particles (100–300, 300–500, 500–700, or 700–900 μm), and microcoil. The TILA-TACE procedure would be as follows

1. Superselectively identify all tumor feeding arteries.
2. Estimate roughly the tumor volume covered by each tumor feeding artery. If there are 3 tumor feeding arteries, divide bicarbonate, doxorubicin-lipiodol emulsion, oxaliplatin, homocamptothecin into 3 parts. If they cover nearly the same volume of tumor, then divide the agents into 3 equal parts. If this is the case, the procedures will be as below:
3. In each tumor feeding artery, about 25 ml bicarbonate would be injected into tumor feeding artery.
4. Then, 3 ml doxorubicin-lipiodol emulsion, 1 ml oxaliplatin, 1 ml homocamptothecin, 3 ml bicarbonate, 20–40 PVA particles (PVA particle size would be chosen according to the diameter of artery), are sequentially injected.
5. Repeat step 4 until the tumor supported by this artery is totally filled (lipiodol oil injection was monitored under fluoroscopic guidance). This cycle may repeat several times.
6. Embolize the tumor feeding artery using PVA of suitable size to block blood stream. Microcoil is used to prevent washout of lipiodol by blood stream, if the diameter of the tumor feeding artery is large, e.g., the internal diameter is larger than 2 mm.
7. The fully occlusion of feeding artery is confirmed by DSA angiography. The deposit of lipiodol oil was assessed by C-arm CT.
8. Repeat step 3 to 7 on the second and the third feeding artery.

cTACE was performed the same as above, except no bicarbonate.

Retreatment was based on the evidence of viable tumor residues. The average sessions of treatment were 4 (range 1–13).

## Adverse events

The following adverse events which might occur during and after TACE procedure were monitored: blood pressure and oxygen saturation during TACE, pain, fever, and signs of liver decompensation after treatment, and biliary system.

## Statistical analysis

Differences in viable tumor residues after the first treatment (the primary endpoint of the study) between the two treatment groups cTACE and TILA-TACE were examined using unpaired *t*-tests and were further adjusted for viable tumor volume before treatment and other potential confounding factors using the general linear model. Log-transformation was conducted to normalize the distribution of viable tumor residues and viable tumor volume before treatment in parametric analyses, with assigning 1 to viable tumor residues if the measured value (ranging 0–216.5%) was 0. We also categorized tumor responses to treatment into four categories according to EASL criteria. Differences in the distribution of categories of tumor responses to treatment between two treatment groups were examined using the proportional odds model after adjustment for viable tumor volume before treatment. Overall survival time was calculated from the date of the first treatment to the date of death from any cause or the last follow-up visit (October 31, 2015). Distributions of overall survival were charted by Kaplan-Meier method and compared with the log-rank test. The overall survival in the RCT was assessed using both the intent-to-treat and per-protocol methods. A two-sided alpha of 0.05 was used for all tests.

## Acknowledgements

This work has been supported in part by the China National 973 project (2013CB911303), China Natural Sciences Foundation projects (81272456, 81470126, 81301707) and the Fundamental Research Funds for the Central Universities, National Ministry of Education, China. The funders have no role in study design, in the collection, analysis, and interpretation of data, in the writing of the report, and in the decision to submit the paper for publication. We thank professors Guo-Hua Fong (University of Connecticut School of Medicine), Mingliang He (The Chinese University of Hong Kong), for critical reading of this manuscript and constructive comments.

## Additional information

### Funding

| Funder | Grant reference number | Author |
|---|---|---|
| National Natural Science Foundation of China | 81301707 | Hao Wu |
| China National 973 Project | 2013CB911303 | Xun Hu |
| National Natural Science Foundation of China | 81272456 | Xun Hu |
| Ministry of Education of the People's Republic of China | Fundamental Research Funds for the Central Universities | Xun Hu |
| National Natural Science Foundation of China | 81470126 | Xun Hu |

The funders had no role in study design, data collection and interpretation, or the decision to submit the work for publication.

### Author contributions

Ming Chao, Conceived the conception, Designed the study, Performed TACE, Wrote the paper, Critical revision, Analysis and interpretation of data, Contributed unpublished essential data or reagents; Hao Wu, Conceived the conception, Designed the study, Analyzed the data, Wrote the paper, Critical revision, Contributed unpublished essential data or reagents; Kai Jin, Performed

TACE, Analyzed the data, Critical revision, Contributed unpublished essential data or reagents; Bin Li, Jianjun Wu, Guangqiang Zhang, Performed TACE, Analysis and interpretation of data, Contributed unpublished essential data or reagents; Gong Yang, Critical revision, Analyzed the data; Xun Hu, Conceived the conception, Designed the study, Wrote the paper, Analyzed the data, Critical revision, Contributed unpublished essential data or reagents

## Author ORCIDs
Xun Hu (iD) http://orcid.org/0000-0002-3784-371X

## Ethics
Clinical trial registration registration number: ChiCTR-IOR-14005319 in Chinese Clinical Trial Registry; protocol can be assessed at http://www.medresman.org/; available as Supplementary file 5). Human subjects: The study was performed with patients' written informed consent and with the approval of hospital's Institutional Review Board (The Second Affiliated Hospital, Zhejiang University School of Medicine).

## Decision letter and Author response
Decision letter https://doi.org/10.7554/eLife.15691.sa1
Author response https://doi.org/10.7554/eLife.15691.sa2

## Additional files

### Supplementary files
• Supplementary file 1. Patient demographics, diagnosis, treatment, and follow-up in cTACE group in the nonrandomized study.

• Supplementary file 2. Patient demographics, diagnosis, treatment, and follow-up in TILA-TACE group in the nonrandomized study.

• Supplementary file 3. Patient demographics, diagnosis, treatment, and follow-up in RCT.

• Supplementary file 4. Sample size estimation for the randomized study.

• Supplementary file 5. Randomized controlled study of bicarbonate-enhanced and conventional transarterial chemoembolization in treatment of hepatocellular carcinoma (protocol number ZUSAHZUCI201401).

• Reporting standard 1. CONSORT flor diagram.

• Reporting standard 2. CONSORT 2010 checklist.

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
