## [Decision Letter]

Thank you for submitting your article "Intratumoral lactic acidosis is a promising therapeutic target for cancer therapy" for consideration by *eLife*. Your article has been favorably evaluated by Charles Sawyers as the Senior editor and four reviewers, including Robert Gatenby and Chi Dang, who is a member of our Board of Reviewing Editors.

The reviewers have discussed the reviews with one another and the Reviewing Editor has drafted this decision to help you prepare a revised submission.

Summary:

In this manuscript, the authors report that local bicarbonate infusion prior to transarterial chemoembolization (TACE) of large hepatocellular carcinomas (HCCs) resulted in remarkable improvement in tumor responses in a nonrandomized cohort of 57 patients followed by a randomized control 20 patients, which supported the findings of the nonrandomized study. Specifically, the clinical responses with bicarbonate were remarkable with early signs of improved survival for patients with large HCCs. The authors speculated that reducing or "targeting" intra-tumoral lactic acidosis may improve the clinical outcome and reported an objective response rate of 100% in the therapy group vs. 44% in the control group. The authors conclude that bicarbonate local infusion markedly enhances the anticancer activity of TACE. If validated, this approach could be a significant improvement over current TACE and support the notion that cancer metabolism, and in this case – acidity, could be manipulated in the clinic. While the data are of significant interest, there are some major concerns that must be addressed.

Essential revisions:

1) The authors should provide additional details on the clinical protocol for administration of bicarbonate and subsequent chemoembolization so that others could reproduce the study. It is not clear whether and how the area and timing of the "local bicarbonate" injection and TACE overlaps to account for the therapeutic advantages. Since both local bicarbonate injection and TACE are both local treatments, how these two local treatments are related to each other are critical information that are missing from this manuscript.

2) In the TACE group, were the patients treated with a placebo, such as saline injection to control for the placebo or procedure-related events that contribute to the differences in outcomes?

3) The control arm with TACE alone achieved a survival of only 14 months, which is far below the expected survival outcomes available in the literature and various treatment guidelines. This may be due to the fact that both BCLC B and C patients were enrolled, which by itself is adding confusion to the interpretation of the data.

4) The authors report a 100% response rate using EASL. If this was the case, the expected survival should really be much greater in the experimental arm. The authors do not make any comments on this fact. In addition, they do not report whether the tumors recurred or whether the patients had to be re-treated. These are critically important clinical questions that the authors must address. The sample sizes appear small to reach confident conclusions. Additional supplemental data files on the individual patients would add significantly and document the data in more detail. The statistical analysis needs much more details in terms of the local control benefit and how to correct for the patients' cross-over between the two treatment groups.

5) Given the complexity of the sample recruitment and group changing during the trial, CONSORT Flow Diagram and checklist will be important and helpful to make the data more transparent and interpretable.

---

## [Author Response]

Essential revisions:1) The authors should provide additional details on the clinical protocol for administration of bicarbonate and subsequent chemoembolization so that others could reproduce the study. It is not clear whether and how the area and timing of the "local bicarbonate" injection and TACE overlaps to account for the therapeutic advantages. Since both local bicarbonate injection and TACE are both local treatments, how these two local treatments are related to each other are critical information that are missing from this manuscript.

We added a detailed protocol of TILA-TACE in the Methods section:

“The following example may give a clearer description of the procedure:

If a tumor (10 cm) is to be treated, according to our experience, we would prepare 150 ml 5% bicarbonate, 60 ml lipiodol doxorubicin emulsion (40 mg doxorubicin dissolved in 30 ml contrast medium and mixed with 30 ml lipiodol), oxaliplatin 150 mg in 20 ml 5% glucose, 40 mg homocamptothecin in 20 ml saline, PVA particles (100-300, 300-500, or 500-700,700-900 μm), and microcoil.

[…]

8. Repeat step 3 to 7 on the second and the third tumor feeding artery.”

2) In the TACE group, were the patients treated with a placebo, such as saline injection to control for the placebo or procedure-related events that contribute to the differences in outcomes?

We agree with the reviewers’ comment that the control group should have been treated with placebo (saline), but in this case, we did not use saline as placebo, because saline injection will change our standard cTACE protocol that we performed for years. If we add saline into the procedure, then the comparison would not be between TILA-TACE and standard cTACE.

In addition to the mechanism by which bicarbonate enhances the anticancer activity of TACE described in our manuscript, there is another mechanism that bicarbonate could enhance the anticancer activity of TACE. As the weakly basic doxorubicin is more toxic in basic pH than in acidic pH condition, intratumoral alkalization by bicarbonate would potentiate the cytotoxicity of doxorubicin. This is a critical point that should be clarified. Therefore, we initiated another small randomized controlled study (Title: Randomized controlled study of bicarbonate-enhanced transarterial chemoembolization and transarterial embolization in treatment of hepatocellular carcinoma; registration number ChiCTR-IPR-15006025 in Chinese Clinical Trial Registry), in which, patients were randomly assigned to TILA-TACE or TILA-TAE (transarterial embolization, the same as TILA-TACE, except no chemotherapeutic agents). So far, 36 patients have been treated, and the objective tumor response rates between TILA-TACE and TILA-TAE group were virtually the same. The study is ongoing, but the results achieved so far strongly suggest that bicarbonate is the major agent that results in massive death of HCC, because chemotherapeutic agents (with or without) did not make a significant difference (100% objective response rate (ORR) of the targeted tumors in the TILA-TACE arm versus 93% ORR of the targeted tumors in the TILA-TAE arm).

Although the above evidence cannot directly answer the question as to whether the same amount of saline injection would exert a similar effect to bicarbonate, it demonstrated that local bicarbonate injection into HCC is effective in control of the tumor.

3) The control arm with TACE alone achieved a survival of only 14 months, which is far below the expected survival outcomes available in the literature and various treatment guidelines. This may be due to the fact that both BCLC B and C patients were enrolled, which by itself is adding confusion to the interpretation of the data.

The control arm with cTACE treatment achieved a median survival of 14 months. At the first glance, the median survival seems short. However, after close analysis, this median survival is most probably not below the expected survival outcome, if the patients’ disease characteristics, particularly BCLC staging and tumor size, are taken into consideration. In this control arm, 27 HCC patients included 9 BCLC C and 18 BCLC B, among which there were 15 patients with tumor size larger than 10 cm and 3 patients with tumor size close to 10 cm (9.4, 9.1, and 8.6 cm). We did not find a paper that reported survival of patients with disease characteristics comparable to the patients in our study, but we may take the reported survival data of patients with different tumor stages and tumor size as references, as detailed below:

Overall survival wise, TACE is more effective for patients with less tumor burden and higher liver reserve than those with higher tumor burden and lower liver reserve (Sieghart et al., 2015; Llovet et al., 2002; Lo et al., 2002; Malagari et al., 2012; Burrel et al., 2012; Takayasu et al., 2012) e.g., the 3-year survival rates for ideal TACE candidates (low tumor burden at BCLC stage A or BCLC stage B) could reach 55%-65% (Malagari et al, 2012; Burrel et al., 2012; Takayasu et al., 2012) whereas the 3-year survival rates for less rigorously selected patients were only 26 – 29% (Llovet et al., 2002; Lo et al., 2002). The 3 year survival of the control arm in this current study was 25.9%.

Tumor size appears to be a major factor to affect survival. LIovet et al. (2002) reported a 30 months median survival of patients at BCLC B and C with tumor size between 4.0 – 5.8 cm; Lo et al. (2002) reported a 17 months median survival of patients at BCLC B and C with tumor size between 4.0 – 14 cm; Huang et al. (2006) Jing-Huan Li (2015), and Paul et al. (2011) reported 9-10 months median survival of patients at BCLC B and C with tumor size larger than 10 cm.

For HCC patients at BCLC C stage, the recommended treatment is sorafenib. Current situation for patients with advanced HCC is desperate. Sorafenib is recommended for treating patients of BCLC C category. Two landmark studies confirmed that sorafenib could prolong modestly the median survival of patients in comparison to placebo (Cheng et al., 2009; Llovet et al., 2008). Cheng et al. reported a median survival of 6.5 months (sorafenib arm) versus 4.2 (placebo), and LIovet et al. reported a median survival of 10.7 months (sorafenib arm) and 7.9 months (placebo). The median survival of 10.7 months in the sorafenib arm was reproduced by several studies (Cheng et al., 2013; Cainap et al., 2015; Johnson et al., 2013).

If the references listed above and the disease characteristics of the patients in our study were taken into consideration, the median survival time of the patients in the control arm of the current study may be not worse than the reported ones.

References:

1) Llovet JM, Real MI, Montana X, Planas R, Coll S, Aponte J, et al. Arterial embolisation or chemoembolisation versus symptomatic treatment in patients with unresectable hepatocellular carcinoma: a randomised controlled trial. Lancet. 2002; 359(9319): 1734-9.

2) Lo CM, Ngan H, Tso WK, Liu CL, Lam CM, Poon RT, et al. Randomized controlled trial of transarterial lipiodol chemoembolization for unresectable hepatocellular carcinoma. Hepatology. 2002; 35(5): 1164-71.

3) Malagari K, Pomoni M, Moschouris H, Bouma E, Koskinas J, Stefaniotou A, et al. Chemoembolization with doxorubicin-eluting beads for unresectable hepatocellular carcinoma: five-year survival analysis. Cardiovasc Intervent Radiol. 2012; 35(5): 1119-28.

4) Burrel M, Reig M, Forner A, Barrufet M, de Lope CR, Tremosini S, et al. Survival of patients with hepatocellular carcinoma treated by transarterial chemoembolisation (TACE) using Drug Eluting Beads. Implications for clinical practice and trial design. J Hepatol. 2012; 56(6): 1330-5.

5) Takayasu K, Arii S, Kudo M, Ichida T, Matsui O, Izumi N, et al. Superselective transarterial chemoembolization for hepatocellular carcinoma. Validation of treatment algorithm proposed by Japanese guidelines. J Hepatol. 2012; 56(4): 886-92.

6) Huang YH, Wu JC, Chen SC, Chen CH, Chiang JH, Huo TI, et al. Survival benefit of transcatheter arterial chemoembolization in patients with hepatocellular carcinoma larger than 10 cm in diameter. Alimentary pharmacology & therapeutics. 2006; 23(1): 129-35.

7) Li JH, Xie XY, Zhang L, Le F, Ge NL, Li LX, et al. Oxaliplatin and 5-fluorouracil hepatic infusion with lipiodolized chemoembolization in large hepatocellular carcinoma. World journal of gastroenterology. 2015; 21(13): 3970-7.

8) Paul SB, Gamanagatti S, Sreenivas V, Chandrashekhara SH, Mukund A, Gulati MS, et al. Trans-arterial chemoembolization (TACE) in patients with unresectable Hepatocellular carcinoma: Experience from a tertiary care centre in India. The Indian journal of radiology & imaging. 2011; 21(2): 113-20.

9) Cheng AL, Kang YK, Chen Z, Tsao CJ, Qin S, Kim JS, et al. Efficacy and safety of sorafenib in patients in the Asia-Pacific region with advanced hepatocellular carcinoma: a phase III randomised, double-blind, placebo-controlled trial. Lancet Oncol. 2009; 10(1): 25-34.

10) Llovet JM, Ricci S, Mazzaferro V, Hilgard P, Gane E, Blanc JF, et al. Sorafenib in advanced hepatocellular carcinoma. N Engl J Med. 2008; 359(4): 378-90.

11) Cheng AL, Kang YK, Lin DY, Park JW, Kudo M, Qin S, et al. Sunitinib versus sorafenib in advanced hepatocellular cancer: results of a randomized phase III trial. Journal of clinical oncology: official journal of the American Society of Clinical Oncology. 2013; 31(32): 4067-75.

12) Cainap C, Qin S, Huang WT, Chung IJ, Pan H, Cheng Y, et al. Linifanib versus Sorafenib in patients with advanced hepatocellular carcinoma: results of a randomized phase III trial. Journal of clinical oncology: official journal of the American Society of Clinical Oncology. 2015; 33(2): 172-9.

13) Johnson PJ, Qin S, Park JW, Poon RT, Raoul JL, Philip PA, et al. Brivanib versus sorafenib as first-line therapy in patients with unresectable, advanced hepatocellular carcinoma: results from the randomized phase III BRISK-FL study. Journal of clinical oncology: official journal of the American Society of Clinical Oncology. 2013; 31(28): 3517-24.

4) The authors report a 100% response rate using EASL. If this was the case, the expected survival should really be much greater in the experimental arm. The authors do not make any comments on this fact.

The 100% ORR in the TILA-TACE group includes CR and PR of the targeted tumors. Many previous studies have confirmed that better local control was an independent prognostic indicator for patient survival (Kim et al., 2015; Shim et al., 2012; Jung et al., 2013; Kim et al., 2013; Riaz et al., 2011; Gillmore et al., 2011; Riaz et al., 2010), and Kim et al. (2015) and Shim et al (2012) further demonstrated that there was a clear prognostic difference between CR, PR, SD, and PD and that PR patients showed a significantly poorer survival than CR patients. Moreover, in the follow up, 16 patients eventually exhibited progressive disease, including 11 patients with new foci in the liver, 1 with new liver foci and lung metastasis, and 3 with lung metastasis, and 1 with bone metastasis, all of which accounted for the death in the experimental arm (Supplementary file 2). These observations also suggest that fast CR and timely control of recurrent tumors in the liver would likely improve the survival of patients, especially those with large tumor burden and low liver reserve. We also mentioned it in the Discussion section of the revised manuscript (third paragraph).

In addition, they do not report whether the tumors recurred or whether the patients had to be re-treated. These are critically important clinical questions that the authors must address.

In the follow up, 16 patients ultimately exhibited progressive disease, including 11 patients with new foci in the liver, 1 with new liver foci and lung metastasis, and 3 with lung metastasis, and 1 with bone metastasis. We added the information into the revised manuscript (Discussion section, third paragraph, and Supplementary files 1–3).

In the Methods section, we added “Retreatment was based on the evidence of viable tumor residues. The average sessions of treatment were 4 (range 1-13)”.

The sample sizes appear small to reach confident conclusions. Additional supplemental data files on the individual patients would add significantly and document the data in more detail.

We added Supplementary files 1–3, which provide detailed information of the patients.

The statistical analysis needs much more details in terms of the local control benefit and how to correct for the patients' cross-over between the two treatment groups.

In the non-randomized cohort of the study, patients recruited in the treatment and control groups differed in some clinicopathological characteristics, although the same inclusion/exclusion criteria were used. In the revised manuscript, we evaluated whether treatment effects (assessed by viable tumor residues, VTR) were confounded by these clinical parameters such as age, BCLC tumor stage, extra-hepatic metastasis, HBV DNA copy numbers, viable tumor volume before treatment, macrovascular invasion, and tumor multifocality using general linear models. Among them, only viable tumor volume before treatment was significantly associated with VTR. We also found a similar association for macrovascular invasion, being of statistically borderline significance. Therefore, we adjusted for these two covariates and estimated multivariable-adjusted VTR using the general linear model. We found that adjustment for these potential confounding factors did not materially alter the results, suggesting that the finding of remarkable improvement in tumor responses in the TILA-TACE vs. cTACE was independent of patients’ clinicopathological characteristics.

We also used the proportional odds model to adjust for viable tumor volume before treatment when comparing differences in the distribution of categories of tumor responses to treatment between two treatment groups.

The overall survival in the RCT was assessed using both the intent-to-treat (ITT) and per-protocal (PP) methods. There was no apparent difference in overall survival between two treatment groups in the ITT analysis. However, a survival advantage appears in TILA-TACE treatment over cTACE treatment in the PP analysis. PP analyses may provide a better sense of the actual biological effect of treatment but are like to introduce biases to the study because of abolishment of randomization. We acknowledged these limitations of the present small RCT in the revised manuscript. We are planning a large-scale RCT to further examine both targeted/local tumor responses, overall and progression-free survival with bicarbonate TACE in patients with large HCCs.

We added the information into in the revised manuscript (Methods, subsection “Statistical analysis”, and Results section).

5) Given the complexity of the sample recruitment and group changing during the trial, CONSORT Flow Diagram and checklist will be important and helpful to make the data more transparent and interpretable.

We added a CONSORT Flow Diagram (Reporting standard 1) and a CONSORT-checklist (Reporting standard 2).